# National survey to estimate sodium and potassium intake and knowledge attitudes and behaviours towards salt consumption of adults in the Sultanate of Oman

Adhra Al-Mawali,[1,2] Lanfranco D'Elia,[3,4] Sathish Kumar Jayapal,[1] Magdi Morsi,[1] Waleed Nasser Al-Shekaili,[1] Avinash D Pinto,[1] Hilal Al-Kharusi,[1] Zainab Al-Balushi,[1] John Idikula,[1] Ayaman Al-Harrasi,[1] Francesco P Cappuccio [3,5,6]

For numbered affiliations see end of article.

**Correspondence to**
Professor Francesco P Cappuccio;
f.p.cappuccio@warwick.ac.uk

## ABSTRACT

**Objectives** To estimate population sodium and potassium intakes and explore knowledge, attitudes and behaviour (KAB) towards the use of salt in adults in the Sultanate of Oman.

**Design** National cross-sectional population-based survey.

**Setting** Proportional random samples, representative of Omani adults (18 years or older), were obtained from all governorates of the Sultanate of Oman.

**Participants** Five hundred and sixty-nine (193 men, 376 women; 18 years or older) were included in the analysis (response rate 57%). Mean age was 39.4 years (SD 13.1). Participants attended a screening including demographic, anthropometric and physical measurements.

**Primary and secondary outcome measures** We assessed dietary sodium, potassium and creatinine by 24-hour urinary sodium (UNa), potassium (UK) and creatinine (UCr) excretions. We collected KAB by a questionnaire on an electronic tablet.

**Results** Mean UNa was 144.3 (78.8) mmol/day, equivalent to 9.0 g of salt/day and potassium excretion 52.6 (32.6) mmol/day, equivalent to 2.36 g/day, after adjusting for non-urinary losses. Men ate significantly more sodium and potassium than women. Only 22% of the sample had a salt intake below the WHO recommended target of 5 g/day and less than 10% met WHO targets for potassium excretion (>90 mmol/day). While 89.1% of those interviewed knew that consuming too much salt could cause serious health problems and only 6.9% felt they were using too much added salt, one in two participants used always or often salt, salty seasonings or salty sauces in cooking or when preparing food at home.

**Conclusions** In the Sultanate of Oman, salt consumption is higher and potassium consumption lower than recommended by WHO, both in men and in women. The present data provide, for the first time, evidence to support a national programme of population salt reduction to prevent the increasing burden of cardiovascular disease in the area.

## Strengths and limitations of this study

► National survey of Omani men and women using 24-hour urine collections.
► Adoption of quality control process to minimise the use of incomplete urine collections.
► Overall response rate was 57%, comparable with other similar population surveys.
► Non-responders did not differ in their baseline characteristics from responders.
► We cannot rule out the risk of selection bias.

## INTRODUCTION

Non-communicable diseases (NCDs) are the leading, yet preventable, causes of death worldwide.[1] The reduction of its burden is now a global health priority of the UN,[2] endorsed by the WHO Action Plan that has identified a set of cost-effective policy options ('best buys'), of which reduction in population salt consumption is one.[3]

In the Sultanate of Oman, NCDs are among the leading causes of death, accounting for 72% of all deaths.[4] Cardiovascular disease (CVD) represents an increasingly common cause of population morbidity and mortality, accounting for 36% of all deaths.[4] It represents a major public health challenge undermining socioeconomic development.[5]

High blood pressure (BP) and unhealthy diets are the leading risk factors for CVD in the world.[1] Raised BP is a determinant of the CVD risk in the Sultanate of Oman, where the prevalence of raised BP in people aged 18 years or older is 33%, higher in men (39%) than in women (27%).[5 6]

High salt (ie, sodium chloride, 1 g=17.1 mmol of sodium) consumption is an important determinant of high BP. A high salt

intake is associated with raised BP that leads to increased risk of vascular diseases.[7–10] In addition, high salt intake is related to adverse health effects independent of its effects on BP.[11–13] A moderate reduction in salt consumption reduces BP[7 8] and it can improve the health outcomes and indirectly reduce the overall mortality through beneficial effect on the BP.[9 10]

The WHO recommends that adults should consume no more than 5 g of salt daily.[14] However, mean daily intakes of salt in most of the countries in the world exceed this recommendation.[15 16] While there is no definitive estimate of population dietary salt intake in the Sultanate of Oman, average consumption could be high, similar to some countries in the subregion.[17 18] In the Sultanate of Oman, it is a common habit to add salt and salty condiments to food at the table and while cooking. Also, the habit of eating out is increasing (especially in urban areas), which leads to an increased salt intake, since restaurants tend to use higher amounts of salt to render food tastier. Our study was designed to support the salt reduction strategy of the Eastern Mediterranean region (EMRO), including the Sultanate of Oman, in which monitoring population salt consumption is one of the three pillars.[19] Current national initiatives include establishment of a multi-sectoral national committee, legislation on salt reduction, development of salt content benchmarks, dietary guidelines.[18] The 'Health Vision 2050' for the Sultanate of Oman was also developed as a roadmap by analysing extensively the status of the Omani health system, the morbidity and mortality in the population, the challenges facing the health system, the expected future developments and changes in the population including macro-social and macroeconomic changes in order to augment the performance of the health system.

In contrast to sodium, epidemiological and intervention studies suggest beneficial effects of dietary potassium on BP and cardiovascular health.[20–22] The Sultanate of Oman lacks data on actual potassium consumption. The WHO currently recommends that adults should consume not less than 90 mmol of potassium daily.[23] Hence, we need reliable data on sodium and potassium intake in the Sultanate of Oman.

The primary aim of the present study was to establish current baseline average consumption of sodium and potassium by 24-hour urine collection, in a national random sample of Omani men and women. The study also aimed to explore knowledge, attitudes and behaviour (KAB) towards dietary salt.

## MATERIAL AND METHODS

### Participants and recruitment

We nested the salt survey within the main Oman NCD survey of 6833 households (online supplemental material 1, text S1). We recruited only one member per household. We designed the salt survey to collect 24-hour urinary samples from a subgroup of at least 90 participants from each governorate (region). The survey included only

**Table 1** Geographical sampling from the Sultanate of Oman

| Governorate | Valid 24-hour urine collections | % |
|---|---|---|
| Muscat | 67 | 11.8 |
| Dhofar | 79 | 13.9 |
| Al-Dakhlia | 45 | 7.9 |
| North Sharqiah | 36 | 6.3 |
| South Sharqiah | 45 | 7.9 |
| North Batina | 81 | 14.2 |
| South Batina | 53 | 9.3 |
| Al-Dhahirah | 46 | 8.1 |
| Al Buraymi | 84 | 14.8 |
| Musandam | 9 | 1.6 |
| Al-Wasta | 24 | 4.2 |
| Total | 569 | 100.0 |

Omani citizens. We included a total of 999 randomly selected Omani men and women. They were all aged 18 years or older. They comprised residents of all governorates of the Sultanate of Oman (table 1). The sample was representative of the national sample for its general characteristics (see online supplemental table 1).

From the sampling frame and according to the EMRO-WHO Protocol,[24] we excluded the following groups: people unable to provide informed consent, those with known history of heart or kidney failure, stroke, liver disease, those who recently began therapy with diuretics (less than 2 weeks), pregnant women, any other conditions that would make 24-hour urine collection difficult. To detect approximately 1 g reduction in salt intake over time using 24-hour urinary sodium excretion (difference ~20 mmol/24 hours), with an SD of 75 mmol/day (alpha=0.05, power=0.80), a minimum sample of 120 individuals per stratum is recommended.[24] Thus, we estimated a minimum recommended sample size of 240 per age and sex groups and adjusted for an anticipated non-response rate of 50%.[24] We stratified the population in groups by sex (men and women). Therefore, 480 individuals were originally needed to be selected (total n=120×2 groups/0.5 attrition=480).

The survey took place between December 2017 and May 2018. From the 999 individuals interviewed in the sampling frame, 569 of them (57.0%) provided suitable data for inclusion in the survey analysis. The general characteristics of the included participants did not differ substantially from those of the excluded participants (see online supplemental table 2). Originally, 262 (26.0%) did not provide complete urine collections (either declaring missing more than one void or providing collections <23 hour or above 25 hours), 87 (8.7%) had missing data, 48 (4.8%) provided urine collections with volume less than 500 mL (conventionally taken as not plausible) and 24 (2.4%) had urinary creatinine excretion outside two

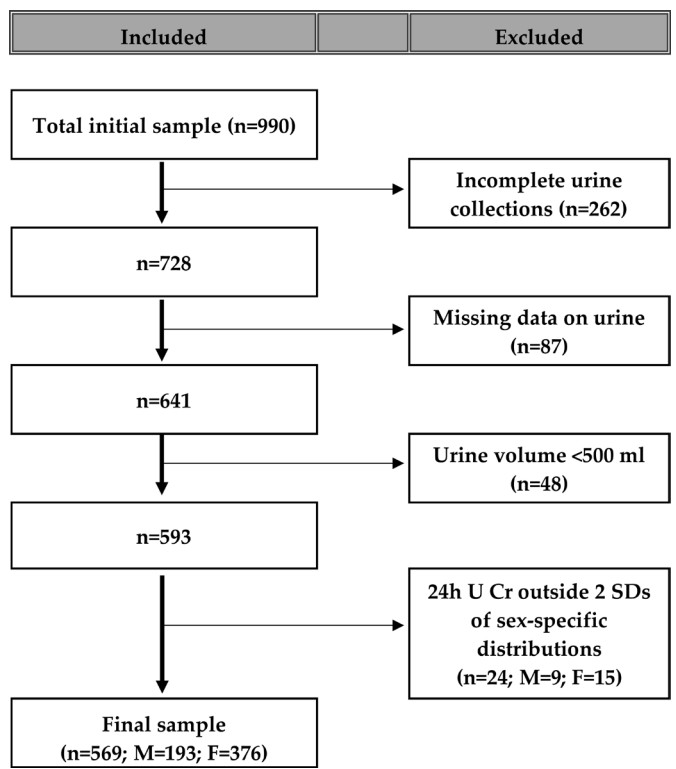

| Included | Excluded |
|---|---|

Total initial sample (n=990)

Incomplete urine collections (n=262)

n=728

Missing data on urine (n=87)

n=641

Urine volume <500 ml (n=48)

n=593

24h U Cr outside 2 SDs of sex-specific distributions (n=24; M=9; F=15)

Final sample (n=569; M=193; F=376)

**Figure 1** Stepwise procedure for the selection of valid participants according to protocol adherence, quality control and completeness of 24-hour urine collections.

SDs of the sex-specific distribution of urinary creatinine in the sample (figure 1).

### Ethical considerations
We carried out the survey in accordance with the Declaration of Helsinki and Good Clinical Practice.[25]

### Patient and public involvement
No patient involved.

### Data collection
We performed the examination in a quiet and comfortable room, with the participants who did not smoke, exercise, eat and consume caffeine before attending and had been instructed to present with a full bladder 30 min before measurements to reduce the risk of underestimating the urine collection. We carried out the survey in three steps: (a) questionnaire survey, (b) physical measurements and (c) 24-hour urine collections.

We based the questionnaire on the Core and Expanded version of the WHO STEPS Instrument for Chronic Disease Risk Factor Surveillance (V.3.0)[26] and country-specific requirements. It contained 11 core, 1 optional and 4 country-specific modules that included a total of 420 questions, to determine sociodemographic characteristics of participants, key behavioural risk factors (tobacco use, harmful alcohol consumption, diet with frequency of fruit and vegetable consumption, high dietary salt consumption, oil and fat use, physical inactivity), knowledge attitudes and behaviour on dietary salt, given lifestyle advises

and additional health-related information not presented here. KABs towards the consumption of salt were assessed by asking participants about the frequency, quantity and type of salt used in the household as well as their cooking habits and their attitudes and perceptions towards dietary salt intake. Processed food was defined, per WHO STEPS protocol, as foods altered from their natural state, such as packaged salty snacks, canned salty food, cheese and processed meat along with country-specific pictorial show cards.

We measured anthropometric indices, BP and heart rate in all participants. Height was in centimetre and body weight in kilogram using a standardised and calibrated SECA813 digital floor scales and 213 portable stadiometers, respectively. Body mass index (BMI) was calculated as weight (kg) divided by height squared ($m^2$). Waist and hip circumferences were measured by a non-stretch SECA201 measuring tape to the nearest millimetre.[24] We took systolic and diastolic BP and heart rate measurements three times in the right arm on a sitting position, using an appropriate cuff and a validated digital device (OMRON M3). We ignored the first measurement and used the mean of second and third measurements for analysis. We took measurements after the participant had rested for 15 min and each with 3 min of rest between the measurements (maximum deviation of cuff pressure measurement ±3 mm Hg and of pulse rate display ±5%). Hypertension is defined as systolic and/or diastolic BP ≥140/90 mm Hg or regular antihypertensive treatment.[27] We obtained a single 24-hour urine collection from the participants. We gave each participant a leaflet with explanations along with the necessary equipment and a record sheet on which they noted the start and the finish times of their urine collection, any missed urine aliquots and any medication taken during the collection. We instructed the participants carefully on urine collection methodology.[24] In an effort to minimise bias, we also requested participants not to change their diet before or during the day of the urine collection. They discarded the first void on waking on the day of collection. Participants then filled the 24-hour urine container over the 24-hour period. On the following day, the field team members visited the household, measured total volume, mixed it thoroughly and obtained a urine sample, which was kept in a cool box for transport to the respective laboratory. On arrival at the laboratory, we either carried out sodium, potassium and creatinine determinations immediately or stored samples in the fridge until the determination (as soon as possible). Sodium and potassium concentration in the urine samples were determined using an ion-selective electrode with an Abott C8000 & Roche Cobas 6000 and expressed in mmol/L.[28] Creatinine concentration was determined through either the kinetic (Abbott C8000) or enzymatic (Roche Cobas 6000) method and expressed in mg/dL.[29] These determinations were carried out in one reference laboratory in each of the 11 governorates, except for two regions (Dhofar and Musandam) which

had two receiving reference laboratories each. All laboratories underwent joint quality control.

## Statistical analysis

We performed all statistical analyses using the SPSS software, V.20 (SPSS). We used t-test for unpaired samples to assess differences between group means and Pearson $\chi^2$ test to test the association between categorical variables. To convert urinary output into dietary intake, we first converted the urinary excretion of sodium (UNa) or potassium (UK) values (mmoL/day) into mg/day (for sodium 1 mmol=23 mg of sodium, for potassium 1 mmol=39 mg). We then multiplied the sodium value by 2.542 to convert dietary sodium (Na) intake into salt (NaCl) intake. We finally multiplied sodium values by 1.05 (assuming that approximately 95% of sodium ingested is excreted).[30] We calculated potassium dietary intake assuming that 85% of the potassium ingested is excreted in the urine.[31] The results were reported as mean (SD), median (IQ range) or as percentages, as appropriate. We considered two-sided p below 0.05 as statistically significant.

## RESULTS

The final population sample included 569 participants between 18 and 69 years old (n=193 or 34% men and n=376 or 66% women), recruited nationally.

## Characteristics of the participants

Table 2 shows the characteristics of the participants. There was no statistically significant difference in the mean age and in BMI between men and women, however, men had significantly higher BP and slower heart rate than women had. The prevalence of hypertension was on average of 27.4%, significantly higher in men than in women (38.5% vs 21.7%, p<0.001).

## Daily urinary excretions of volume, sodium, potassium and creatinine and salt and potassium intake

Average urinary volume excretion was 1354 mL/day, being higher in men than women (table 3). Average urinary creatinine excretion was 1.33 g/day, being again higher in men than women (table 3). Mean urinary sodium was 144.3 (SD 78.8) mmol/24 hours, equivalent to a mean consumption of 9.0 (4.9) g of salt per day (table 3). Men excreted more sodium than women did (mean difference 15.0 mmol/24 hours, p<0.05), equivalent to ~1.0 g of higher salt consumption than women did. Only 22% of the participants met the levels of salt intake of 5 g or less recommended by the WHO, with no difference between sexes. Mean urinary potassium was 52.6 (32.6) mmol/24 hours, equivalent to a mean consumption of 2.36 (1.46) g of potassium per day (table 3).

Men excreted significantly more potassium than women; 9.1% of participants met the levels of potassium intake of 90 mmol/day or more recommended by the WHO.

The sex difference in total daily salt and potassium intakes is almost entirely due to the fact that men eat more food than women, as they are taller and heavier, despite having comparable BMI. This is a consistent finding across populations in different countries and from different continents.

## KABs towards salt intake and other eating patterns.

KABs towards the consumption of salt are presented in table 4. A total of 28.1% of respondents mentioned that

| Table 2 Characteristics of the participants | | | |
|---|---|---|---|
| Variable | All (n=569) | Men (n=193) | Women (n=376) |
| Age (years) | 39.4 (13.1) | 38.7 (14.3) | 39.8 (12.5) |
| Height (cm)† | 159.4 (11.2) | 167.9 (9.7) | 154.9 (9.2)‡ |
| Weight (kg)† | 74.9 (21.5) | 81.4 (22.5) | 71.4 (20.1)‡ |
| Body mass index (kg/m²)† | 29.3 (7.2) | 28.9 (7.6) | 29.5 (7.0) |
| Waist circumference (cm)† | 93.8 (15.7) | 95.0 (15.7) | 93.2 (15.7) |
| Hip circumference (cm)† | 104.5 (15.0) | 102.6 (13.7) | 105.5 (15.6)§ |
| Systolic blood pressure (mm Hg)* | 125.9 (18.2) | 134.0 (17.0) | 121.7 (17.3)‡ |
| Diastolic blood pressure (mm Hg)* | 80.9 (10.7) | 83.4 (11.5) | 79.7 (10.1)‡ |
| Pulse rate (b/min)* | 79.8 (10.5) | 78.5 (11.8) | 80.4 (9.8)§ |
| Hypertension N (%)* | 175 (30.8) | 77 (39.9) | 98 (26.1)‡ |
| On antihypertensives N (%)* | 50 (28.6) | 15 (19.5) | 35 (35.7)§ |

Results are mean (SD) or N(%).
Hypertension: Systolic Blood Pressure / Diastolic Blood Pressure >140/90 mm Hg or on current therapy for high blood pressure.
*3 missing values (1m, 2w) (0.5%).
†18 missing values (1m, 17w) (4%).
‡p<0.01.
§p<0.05 when compared with men.

**Table 3** Daily urinary excretions of volume, sodium, potassium and creatinine, estimates of salt and potassium intake and proportion of participants meeting WHO recommended targets for salt and potassium consumption.

| Variable | All (n=569) | Men (n=193) | Women (n=376) |
|---|---|---|---|
| Volume (mL/24 hours) | 1354 (725) | 1392 (712) | 1335 (731)* |
| | 1129 (855–1618) | 1150 (900–1721) | 1122 (827–1593) |
| Sodium (mmol/24 hours) | 144.3 (78.8) | 154.2 (87.4) | 139.2 (73.6)* |
| | 129.6 (85.7–187.4) | 135.4 (87.1–204.8) | 126.8 (83.9–179.4) |
| Potassium (mmol/24 hours) | 52.6 (32.6) | 56.4 (32.4) | 50.6 (32.5)* |
| | 46.4 (31.4–64.9) | 50.9 (33.8–73.2) | 44.7 (30.2–61.6) |
| Sodium-to-potassium ratio | 3.3 (3.4) | 3.5 (4.2) | 3.2 (2.8) |
| | 2.8 (2.0–3.8) | 2.7 (1.9–3.9) | 2.8 (2.1–3.8) |
| Creatinine (g/24 hours) | 1.33 (0.71) | 1.72 (0.87) | 1.13 (0.52)‡ |
| | 1.18 (0.86–1.63) | 1.61 (1.16–2.12) | 1.02 (0.81–1.36) |
| Salt intake (g/day) | 9.0 (4.9) | 9.6 (5.5) | 8.7 (4.6)* |
| | 8.1 (5.3–11.7) | 8.5 (5.4–12.8) | 7.9 (5.2–11.2) |
| Potassium intake (g/day) | 2.36 (1.46) | 2.53 (1.45) | 2.27 (1.46)* |
| | 2.08 (1.41–2.91) | 2.28 (1.52–3.28) | 2.00 (1.35–2.76) |
| Salt <5 g/day N (%) | 124 (21.8) | 40 (20.7) | 84 (22.3) |
| Potassium >90 mmol/day N (%) | 52 (9.1) | 24 (12.4) | 28 (7.4)† |

Results are mean (SD) and median (25th–75th percentile) or N (%).
*p<0.05.
†p=0.008.
‡p<0.001.

**Table 4** Knowledge, attitudes and behaviours towards salt consumption

| Question | All (n=569) | Men (n=193) | Women (n=376) |
|---|---|---|---|
| How often do you add salt or salty sauces to your food? | | | |
| Often/always | 28.1% | 22.8% | 30.8%* |
| Sometimes | 21.3% | 17.1% | 23.4% |
| Rarely/never | 50.6% | 60.1% | 45.8% |
| How often is salt, salty seasoning or salty sauces added in cooking or preparing food at home?‡ | | | |
| Often/always | 47.0% | 44.8% | 48.1%† |
| Sometimes | 16.5% | 12.5% | 18.6% |
| Rarely/never | 36.5 % | 42.7% | 33.3% |
| How often do you eat processed food?‡ | | | |
| Often/always | 22.3% | 22.8% | 22.1% |
| Sometimes | 38.3% | 35.8% | 39.5% |
| Rarely/never | 39.4% | 41.4% | 38.4% |
| How much salt or salty sauces do you think you consume?‡ | | | |
| Too much/far too much | 6.9% | 7.8% | 6.4% |
| Just the right amount | 66.8% | 61.3% | 69.7% |
| Too little/far too little | 26.3% | 30.9% | 23.9% |
| Do you think that too much salt or salty sauces could cause a serious health problem?‡ | | | |
| Yes | 89.1% | 90.1% | 88.6% |

*Results are percentages p=0.005.
†p=0.04 when compared with men.
‡Reduced numbers due to missing values.

**Table 5** Frequency of other dietary patterns

| Question | All (n=569) | Men (n=193) | Women (n=376) |
|---|---|---|---|
| In a typical week, on how many days do you eat fruit?§ | | | |
| <5 | 32.0% | 40.4% | 27.7%† |
| ≥5 | 68.0% | 59.6% | 72.3% |
| How many servings of fruit do you eat on one of those days?§ | | | |
| <3 | 45.5% | 41.8% | 47.2% |
| ≥3 | 54.5% | 58.2% | 52.8% |
| In a typical week, on how many days do you eat vegetables?§ | | | |
| <5 | 24.1% | 24.9% | 23.7% |
| ≥5 | 75.9% | 75.1% | 76.3% |
| How many servings of vegetables do you eat on one of those days?§ | | | |
| <3 | 59.9% | 65.9% | 56.9%‡ |
| ≥3 | 40.1% | 34.1% | 43.1% |
| What type of oil or fat is most often used for meal preparation in your household?§ | | | |
| Vegetable oil | 90.8% | 91.1% | 90.7% |
| Other (lard, suet, butter, ghee) | 9.0% | 8.9% | 9.0% |
| None used | 0.2% | 0 | 0.3% |
| On average, how many meals per week do you eat that were not prepared at home?§ | | | |
| 0 | 45.8% | 32.1% | 52.8%* |
| ≥1 | 54.2% | 67.9% | 47.2% |

*Results are percentages p<0.0001.
†p=0.002.
‡p=0.04 versus men.
§Reduced numbers due to missing values.

they added salt or salty sauces always or often to food. The percentage of women who added salt or salty sauces always or often to their meal was significantly higher than that of men (30.8% vs 22.8%; p=0.005). A total of 47.0% of respondents reported that they always or often added salt, salty seasonings or sauces when cooking or preparing food at home, women more than men (48.1% vs 44.8%; p=0.04). More than 1 in 5 (22.3%) mentioned that they consumed processed foods high in salt always or often. Very few (6.9%), however, felt they consumed too much salt or salty sauces, although 89.1% knew that consuming too much salt could cause serious health problems. We also asked participants about dietary attitudes about the consumption of fruit and vegetables, oil or fats (table 5). Interestingly, 68.0% consumed fruit at least 5 days a week and 54.5% at least three servings on these days. Men appeared to report more fruit consumption than women did (40.4% vs 27.7%; p=0.002). Vegetables were also consumed frequently (75.9% at least 5 days a week), with 40.1% having at least three servings on one

of those days (women more frequently than men). The majority (90.8%) used vegetable oil for meal preparation in the household and more than half (54.2%) consumed food prepared outside home at least once a week. Men were more likely than women to do so (67.9% vs 47.2%; p<0.001).

## DISCUSSION

This is the first nationally representative population-based survey carried out in the Sultanate of Oman assessing dietary sodium and potassium consumption in adult Omani men and women, using the gold standard measure of 24-hour urinary sodium and potassium excretions as a biomarker of intake. The results show that salt consumption is higher and potassium consumption is lower than recommended by the WHO,[14 23] both in men and women. Average population salt consumption was 9.0 g/day, almost double the WHO recommended maximum population target of 5 g/day.[14] Less than one in four participants met these targets. Salt consumption varied across governorates, being the lowest in South Sharqiah (5.3 g/day) and the highest in Al-Dhahirah (14.8 g/day). Average population excretion of potassium was 53 mmol/day (equivalent to about 2.36 g/day), lower than the WHO recommended maximum population target of >90 mmol/day, equivalent to approximately 3.90 g/day (assuming urinary potassium being 85% of the intake).[23] Potassium consumption also varied across governorates, being the lowest in Al-Wasta (1.41 g/day) and the highest in North Sharqiah (4.25 g/day). The urinary sodium-to-potassium ratio averaged 3.3, with no difference between men and women. Findings from the International Collaborative Study on Salt (INTERSALT) study showed that a difference in sodium-to-potassium ratio from 3.1 to 1.0 was associated with a 3.36 mm Hg difference in population systolic BP.[32 33] The Trial Of Hypertension Prevention (TOHP) study reported a direct association between the urinary sodium-to-potassium ratio and CVD.[10 34 35] Moreover, a unit difference in the ratio would be associated with a 13% reduction in total mortality.[35] Measuring the ratio is obviously important, although no evidence-based global guidelines have determined population targets, as yet.

Our questionnaire revealed that half of the population seen often used sauces and condiments (invariably containing high concentrations of salt) but only 10% believed this was too much. A quarter of the surveyed population added salt to food regularly, one in five ate processed food often and more than half of the population ate out at least once a week, with men more likely than women. These results, in addition to those obtained in previous surveys on unhealthy dietary habits, support the National Health Vision set by the Sultanate of Oman to reduce the burden of CVD.[36] This document sets the health visions for the country in 40 years. The comprehensive analyses of many factors affecting the population health and the healthcare system indicate that NCDs,

in the context of increased life expectancy and population ageing, pose a significant threat to the health of the Omani people and it identifies the need to be able to respond to this challenge. Population salt reduction is one of the priorities.

## Comparison with countries of the Gulf Co-operation Council and of the Arab Peninsula

In the Gulf Co-operation Council (GCC) countries, populations lead a sedentary lifestyle, both hypertension and obesity are common[17] and they are major contributors to NCDs.[37] The estimated total mortality in GCC countries attributable to NCDs varies from 65% to 78%, with the highest estimates in Bahrain and Saudi Arabia and the lowest in Oman and Qatar, respectively.[37] Salt intake is deemed high in most countries of the EMRO Region, although there are only a few studies that directly measured population levels, with inconsistent results due to methodological inadequacies.[17 18] The Global Burden of Disease (GBD) estimates of average salt consumption using a Bayesian model suggest that salt consumption in GCC countries may vary from 8.0 g/day in Saudi Arabia to 13.5 g/day in Bahrain.[38] Estimates of salt intake in neighbouring countries would also range between 7.8 g/day in Lebanon and 10.3 g/day in Jordan.[38] The present study is one of the few nationally representative surveys in GCC countries using the gold standard method of assessment of dietary salt intake. Its results suggest an intake close to that estimated by the GBD. In addition to the GBD, however, our study also provides, for the first time, direct measures of average population potassium consumption also targeted by WHO recommendations for cardiovascular prevention.[20 23]

Many countries of the EMRO Region of WHO are developing and/or implementing national initiatives to decrease population salt intake.[18] National initiatives include the establishment of national multisectoral committees, the engagement of the government through regulatory measures and legislation (Bahrain, Iran, Jordan, Oman, Qatar), the specification of the food categories prioritised for action such as bread (Kuwait, Qatar) and canned foods (Iran), the development of national benchmarks and targets (Bahrain, Iran, Oman), dietary guidelines (Afghanistan, Lebanon, Oman, Saudi Arabia), media awareness campaigns (Lebanon, United Arab Emirates), salt labelling, collaborative actions involving the food industry and/or restaurants and food caterers (Kuwait, Qatar, United Arab Emirates) and the monitoring and evaluation of sodium intakes and salt content of foods (Iran, Lebanon, Oman, Qatar).

## Comparisons with studies in other countries

National salt and potassium consumption surveys have been carried out in almost all regions of the world, especially in response to the recommendations from the WHO that identified population salt reduction as one of the most cost-effective and feasible approaches to prevent NCDs.[2 3] Globally, there is a high variation in the readiness of countries to adopt and implement the different aspects of the overall strategy, with low-income and middle-income countries still lagging behind.[39 40] Nevertheless, where surveys have been carried out to establish the size of the problem, average levels of salt intake have been very high in countries of Eastern Europe (10.8 g/day in Moldova and 11.6 g/day in Montenegro with potassium about 30% lower than recommended and sodium-to-potassium ratios of 3.0 and 2.4, respectively),[41 42] Central Asia (17.2 and 18.8 g/day in two sites of Kazakhstan),[43] China (twofold North-South gradient from 15.6 to 8.4 g/day and potassium about 60% lower than recommended)[44] and Australasia (about 9.0 g/day weighted means in Australia and 11.7 g/day in the Fiji Islands),[45 46] indicating urgent need for population interventions. The same studies have invariably indicated lower than recommended potassium excretion and high sodium-to-potassium ratio. In this respect, the average intake of sodium in Oman seems reassuring, as the achievement of the set targets appears more feasible that in other countries where intake currently still exceeds 10 g/day. However, potassium consumption is nearly half of what is recommended,[23] resulting in a high sodium-to-potassium ratio.

## Strengths and limitations

Our study has several strengths. First, it is a population-based survey across the whole country. Second, it included all adults. Third, it included both men and women. These study characteristics would allow with greater confidence the extrapolation of results to the whole country population, rather than those conducted in selected groups including patients,[47] young female university students[48] or children.[49] Fourth, it used the current preferred methodology for estimating salt consumption. Fifth, we applied a rigorous quality control protocol to ensure completeness of urine collections and to minimise both under and overestimations. Current recommendations suggest the use of single complete 24-hour urine samples, collected from a representative population sample to assess the population's current 24-hour dietary sodium ingestion.[50] The role of single-spot or short duration timed urine collections in assessing population average sodium intake requires more research. Single or multiple spot or short duration timed urine collections are, on the other hand, not recommended for assessing an individual's sodium intake especially in relationship to health outcomes.[50] Twenty-four-hour diet recall and diet records inaccurately measure dietary sodium intake in individuals compared with the gold standard 24-hour urinary excretion.[51] Furthermore, there is poor agreement between estimates of sodium intake from food-frequency questionnaires and 24-hour urine samples.[52] Sixth, it has measured directly the amount of potassium consumption, additional nutrient targeted for cardiovascular prevention.[20 23] Seventh, we standardised fieldwork and used standardised laboratory methodologies across the country. Eighth, all laboratories underwent joint quality control.[25]

There are limitations too. First, we included only 57% of the urine samples originally collected from willing individuals. This was due to the stringent quality control that has led to the exclusion of incomplete or erroneous collections.[24] This could have introduced a self-selection bias. The comparison of the baseline characteristics of the studies sample versus the excluded group suggests that the two groups were comparable for general characteristics, with the exception of the latter being 2 years younger and having a 1.8 mm Hg lower diastolic BP. Second, we assessed urinary sodium and potassium excretions only once. While we cannot characterise an individual's intake in such a way,[50] there is less likelihood of a bias of group estimates. Third, although we requested participants not to change their diet prior to urine collection, it would be difficult to rule out entirely any bias during collection. Fourth, although we administered a questionnaire to derive KABs towards the use of salt, we were unable to establish the relative contribution of discretionary sources of salt and the most common foods contributing to salt as well as potassium consumption.

### Potential impact

The population in the Sultanate of Oman is of just over 5 million (Ministry of Health Annual Health Report, 2018 estimates), of which about 2.3 million are Omani nationals[4] (surveyed in the present study). Approximately 51% are 25 years or older. To meet a 30% reduction in population salt consumption set by WHO by 2025, the Sultanate of Oman should aim at a 2.7 g/day salt reduction nationally. This reduction would avert 8.1% CVD deaths per year and more non-fatal events and disabilities.[2] Additional benefits would be achieved if we increased at the same time population potassium intake towards WHO set targets, leading to a significant reduction in the sodium-to-potassium ratio in the diet. This could be achieved not only by increasing consumption of plant-based foods but also by enriching the diet with potassium-rich salt substitution in food manufacturing and processing or by using potassium-reach salts instead of sodium chloride, where sodium intake is predominantly originating from discretionary sources. Potassium-rich salts lower BP effectively[53] and the potential risk associated with potassium supplementation used at a population level[54] would be offset by a net reduction in CVD deaths.[55]

### Policy implications

The Sultanate of Oman has embraced among its health priorities the prevention and control of NCDs and improvement in nutrition[4] in line with the strategic directions of WHO endorsed by the EMRO in 2012 and 2013[18]. Since then several countries have conducted dietary studies in an attempt to assess the population's salt and potassium intake.[18] Studies in the area have also attempted to identify the major dietary contributors to sodium intake. Studies are still limited and there are large variations in dietary habits in the region due to cultural, ethnic, religious and social heterogeneity. The most common source of salt consumption across the region is bread,[18 56] in all its different forms, with other sources being more relevant in different countries. In Lebanon[57] and Bahrain,[58] dairy products are common sources, while in Morocco,[59] major contributors to salt consumption include cereal-based products, spices and condiments and milk products. These indications, together with the awareness and behaviours measured, suggest that to reduce population salt consumption in the Sultanate of Oman, the following initiatives should be taken: (a) improving salt-related knowledge through health promotion campaigns, (b) measuring major sources of salt consumption, (c) establishing collaborations with local authorities to reduce the amount of salt used in traditional bread making and locally produced condiments, (d) adopting a labelling strategy for imported foods with high salt content. In addition, the Ministry of Health should develop strategies and methodologies to measure the indicators of population salt consumption.[60]

### CONCLUSIONS

This study demonstrates that salt consumption in the Sultanate of Oman is high and should be reduced through a public health action aiming at the entire population. Likewise, potassium consumption is particularly low. The KABs survey indicates areas of limited awareness. Education of the dangers of high salt consumption and where salt is hidden, of the benefits of increasing potassium through fruit, vegetables, nuts and legumes, alongside accurate labelling and marketing of food, surveillance to measure and monitor salt use and reformulating bread are all important elements of an effective national salt reduction programme.[18 19 61]

**Author affiliations**
[1]Centre of Studies & Research, Ministry of Health, Muscat, Oman
[2]The Research Council, Seeb, Oman
[3]WHO Collaborating Centre for Nutrition, University of Warwick, Coventry, UK
[4]Department of Clinical Medicine and Surgery, "Federico II" University of Naples Medical School, Naples, Italy
[5]Division of Health Sciences, Warwick Medical School, University of Warwick, Coventry, UK
[6]Division of Medicine, University Hospitals Coventry & Warwickshire NHS Trust, Coventry, UK

**Acknowledgements** We carried out the present analysis under the terms of reference of the WHO Collaborating Centre for Nutrition at the University of Warwick. The authors would like to express gratitude to the EMRO WHO Regional Office (Dr Ayoub Al-Jawaldeh) and the WHO Country Office of Oman (Dr Ruth Mabry) for facilitating the study, the Department of Food Science and Human Nutrition of Sultan Qaboos University (Dr Lyutha Al-Subhi) and other members of the Ministry of Health (Dr Amel Ibrahim, Dr Ruqaya Balushi and Dr Salima) for their helpful discussions in the preparation of the protocol. Special thanks go to the team of the Ministry of Health of the Sultanate of Oman involved in the preparation of the survey and data collection. Finally, we wish to express our gratitude to His Excellency the Under secretary for Health Affairs of the Sultanate of Oman, Dr Mohammed bin Saif Al Hosni, and his Advisor, Dr Mahmood Shaban, for their advice, endorsements and support at a national level.

**Contributors** FPC developed the study design and protocol, contributed to the analysis and drafted the manuscript, AA-M trained local teams, coordinated quality control and data collection. MM and LD carried out quality control and statistical

analysis. AA-M, SKJ, WNA-S, ADP, HA-K, ZA-B, JI, AA-H coordinated the study, carried out the fieldwork and liaised with the local laboratory. MM helped with the drawing of the stratified random sample from the sampling frame. All authors contributed to the interpretation of the findings and they contributed significantly to the final version of the manuscript. FPC is the guarantor.

**Funding** The Ministry of Health of the Sultanate of Oman and the EMRO Regional Office of the World Health Organization supported the study.

**Competing interests** AA-M, SKJ, MM, WNA-S, ADP, HA-K, ZA-B, JI, AA-H are all staff of the Ministry of Health of the Sultanate of Oman. FPC is a technical advisor to the World Health Organization, unpaid member of Action on Salt and WASH. LD was a technical advisor to the World Health Organization and is a member of the Scientific Committee of the Italian Society of Human Nutrition.

**Patient consent for publication** Not required.

**Ethics approval** We obtained ethical approval for the survey from the Research and Ethics Review and Approval Committee (RERAC) of the Ministry of Health of the Sultanate of Oman and participants provided written informed consent to take part.

**Provenance and peer review** Not commissioned; externally peer reviewed.

**Data availability statement** Data are available upon reasonable request. No individual participant data will be available. Study protocol available in Supplementary Material. Any other data sharing proposal must be submitted in writing to the Ministry of Health of the Sultanate of Oman.

**ORCID iD**
Francesco P Cappuccio http://orcid.org/0000-0002-7842-5493

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
