## [Reviewer comments · BMJ Open]

ARTICLE DETAILS

TITLE (PROVISIONAL)	A national survey to estimate sodium and potassium intake, and knowledge attitudes and behaviours towards salt consumption of adults in the Sultanate of Oman.
AUTHORS	Al-Mawali, Adhra; D'Elia, L; Jayapal, Sathish; Morsi, Magdi; Al-Shekaili, Waleed; Pinto, Avinash; Al-Kharusi, Hilal; Al-Balushi, Zainab; Idikula, John; Al Harrasi, Ayaman; Cappuccio, Francesco Paolo

VERSION 1 – REVIEW

REVIEWER	Joao Breda WHO/Europe, Copenhagen
REVIEW RETURNED	17-Feb-2020

GENERAL COMMENTS	Please compare your results and study attrition with other National studies in other Regions.
---

REVIEWER	Dr Marike Cockeran North-West University South Africa
REVIEW RETURNED	20-Feb-2020

GENERAL COMMENTS	Please note that I have only reviewed the statistical aspects of this manuscript. Page 8, line 33: The output, reported in the tables, is based only on independent t-tests. Therefore, ANOVA can be left out on line 33. Page 8, line 47: The output, reported in the tables, is based only on Mean (SD). Therefore, 95% CI can be left out on line 47. Page 9, lines 31-32: "Urinary potassium excretion showed a normal distribution with a tail skewed to the right (i.e. towards higher values)". This statement is contradictory, if the variable does follow a normal distribution, it cannot also have a tail skewed to the right. Are the mean and standard deviation the correct descriptive statistics to use, if the variable is skewed to the right? Page 19, Table 2: Report hypertension as follows: n (%), similar to Table 3. Page 20, Table 3: I feel it is not necessary to have #p=0.008. Usually, only p<0.01, p<0.05 or p<0.1 are used in the footnote. Page 20, Table 3: To what does "vs men" in the footnote refer to? Page 21, Table 4: To what does "vs men" in the footnote refer to?
--

	Page 22, Table 5: I feel it is not necessary to have #p=0.002. Usually, only p<0.01, p<0.05 or p<0.1 are used in the footnote.
--	--

REVIEWER	Kristy Bolton Deakin University, Australia
REVIEW RETURNED	14-May-2020

GENERAL COMMENTS	Overall comments: Given WHO's aim for member states to reduce salt consumption by 30% by 2025 in order to improve health outcomes at a population level in the most cost effective way, this paper is highly relevant and will inform salt reduction strategies in Oman. However, the methodology currently lacks some specific details – which can be found in comprehensive supplementary file, however I think would be beneficial to be included in the actual paper in brief form. The discussion section could also be expanded and the findings of potassium included, along with acknowledging the importance of the Na:K. Please see comments below. Abstract:  • Objectives could state who the Na, K intake, KABS were measured in whilst this is mentioned in the setting I think the objective could be more specific with this detail– i.e. similar to the title – To estimate population.....in adults living in Oman • Participants: Seems to be a typo – no space between women 18 years or older. Suggest replacing “They attended a” To “Participants attended a...” • Primary and secondary outcome measures: suggest include a word to describe the questionnaire – i.e. hard copy questionnaire or online questionnaire • Results: Were these values adjusted for non-urinary loss of sodium? Did men eat significantly more Na and K than women? It's unclear. • Conclusion: suggest replacing “back up” with “support” or similar Strengths and limitations  • Is the study nationally representative? How? By age and gender? N=569 (193 M, 376F) – is this representative of Oman? What about the spread of age? • What do you mean by adopted a stringent quality control process to minimise the use of incomplete urine collections? I understand word constraints here, but more succinct detail on these quality control processes would be helpful • The last point about selection bias and limitations with generalisability seems to work against point 2 which states this study population is a nationally representative cohort? This is confusing for the reader. What do you mean here? Introduction  • Line 46 – how is the habit of eating out/increasing in urban areas contributing to salt consumption? You could expand here a little. • Line 48: what are the current salt reduction strategies – can you briefly describe these to give the reader an idea of how your work contributes to them? • Line 52: I think you could also discuss the importance of Na:K ratio and benefits for health here, not just focusing solely on K? Materials and methods  • Data collection Line 8: What was the rationale for the participant presenting with a full bladder 30 minutes before measurements? • Data collection line 22: did you collect any medications participants might regularly take?
--

	 • Data collection line 54: the following day after the 24 hour collection, did field team-members visit households for the urine collection, or did the participant take the urine collection somewhere? • Line 31 – you mention the modules that were part of the survey, but how many questions did it include? Was the questionnaire a hard copy or asked verbally? How long did it take the participant to complete the survey? Given the focus in this paper is on KABs, what were the specific survey questions related to KABs? More detail here would be helpful for the reader. You present something similar in Line 43 of the results which might be helpful to be moved here: Knowledge, attitude and behaviours toward the consumption of salt was assessed by asking participants about the frequency, quantity and type of salt used in the household, as well as their cooking habits and their attitudes towards dietary salt • Line 60 onwards – “we either carried out determinations immediately or stored samples in the fridge until the determination (as soon as possible). We determined sodium, potassium, creatinine immediately” – this second sentence is opposite to the first? This is confusing for the reader. • Statistical analysis – I think sample size calculations would be better suited to be presented early – participants per stratum etc – this has not been clearly explained from the outset which is important and relates to your recruitment process earlier described. • Strengths and limitations dot point stated that “the study has adopted a stringent quality control process to minimize the use of incomplete urine collections” – where is the description of these processes in the methods section? • Did you adjust your sample with any post-stratification weights? • Was the length of the urine collection standardised to a 24h period? • Was creatinine used as a measure of urine collection completeness? • What were the exclusion criteria for incomplete urines? E.g. creatinine levels by gender, outliers, missing urine etc. • Results  • Line 58 – how did you define processed foods? As per NOVA classification, or otherwise? This should be stated in your methods section when describing the survey questions • Were there any differences in consumption by socioeconomic status? Discussion:  • The health vision is mentioned in line 56 – given this I think it would be helpful to include some context about what this is here, or in the introduction when you discuss salt reduction interventions • Reasons for differences in salt /potassium consumption by governorates? Do they differ in rurality? Are there differences in local dishes consumed? Any insight the authors can provide to interpret these findings? • RE: comparison with other countries (GCC and Arab Peninsula) – some specific comparison of your study findings compared to other countries would be helpful. Do these countries currently have health visions/salt reduction strategies which might influence population salt consumption? • Given the focus was also KABs, this seems to be missing from the discussion section – how does this compare to similar nations? This would also feed into informing the health vision/direction of intervention strategies?
--	---

	 • Strengths and Limitations: line 36 – how is including all adults a strength of this study? It is a population based survey, but how reflective is your sample by age/gender to national population data? • Line 43: Fifth, we applied a rigorous quality control protocol to ensure completeness of urine collections, and to minimize both under and over-estimations.....there is no detail described about what these processes/protocol was • Limitations: First, we analysed only 57% of the urine samples originally collected from willing individuals. This was due to the stringent quality control that has led to the exclusion of incomplete or erroneous collections.....I thought 57% was the response rate – i.e. number of people consented/ number people invited...is this figure now only the proportion of complete urines collected? This is a little confusing and would benefit from clarification throughout the paper. And how does this introduce self-selection bias? Can you provide a rationale for this? • A potential limitation is bias/changing one's diet prior to the urine collection – whilst acknowledged in the methods section as an instruction not to change diet, this could still be a limitation and worth mentioning here. • Can you acknowledge that without knowing dietary intake, i.e. kJ E consumed in males and females, that males may have higher sodium content due to consuming more food? • Without really understanding how the questionnaire was administered (i.e by an interviewer? On paper? Online?) It is hard to know whether social desirability bias (to the interviewer) or recall bias could influence responses. • I think this data is limited in some respects to informing intervention strategies as diet recalls have not been conducted therefore key sources of dietary salt and potassium aren't known. • RE: policy implications “These indications, together with the awareness and behaviours measured, suggest that policy priorities⁴⁵ to reduce population salt consumption in the Sultanate of Oman would require (a) improvement of salt-related knowledge through health promotion campaigns, (b) assessment of major sources of salt consumption, (c) establishing collaborations with the local authorities to reduce the amount of salt used in traditional bread making and other identified sources like locally produced condiments, (d) adopting a labelling strategy for imported foods with high salt content.” Can you explain what you mean by policy priorities? What sort of policy do you propose might be promising in improving salt related KABs? How will policies assess major sources of salt consumption? Can the authors expand upon this, and draw from existing salt reduction strategies? • Bigger picture implications of this data could be expanded – could this data inform salt reduction strategies? • Potassium levels are not addressed in the discussion – how do you propose you increase potassium consumption to adequate levels? And what about the importance of the Na:K? • Conclusion: lines 31. These are great suggestions for public health promotion, however the data in your study does not appear to directly inform these approaches? • Table 2: missing a key/legend for * • Table 3: *p<0.05 – can you clarify that this analysis was conducted on males vs females? What does +p<0.001 vs men mean? • Table 4: *p<0.05 – can you clarify that this analysis was conducted on males vs females? What does +p=0.04 vs men mean? RE: reduced numbers due to missing values – how many
--	--

	were missing values and why were they missing? Did you conduct an analysis with just complete values and were the same % / effects seen? Same comments for Table 5.
--	---

VERSION 1 – AUTHOR RESPONSE

Reviewer: 1

Please compare your results and study attrition with other National studies in other Regions.

Thank you – we have now added an independent short paragraph in Discussion to acknowledge the global action on determining populations' salt consumption. We have also added a few selected references (#35-#42).

Reviewer: 2

1. Page 8, line 33: The output, reported in the tables, is based only on independent t-tests. Therefore, ANOVA can be left out on line 33.

Done

2. Page 8, line 47: The output, reported in the tables, is based only on Mean (SD). Therefore, 95% CI can be left out on line 47.

Done

3. Page 9, lines 31-32: "Urinary potassium excretion showed a normal distribution with a tail skewed to the right (i.e. towards higher values)". This statement is contradictory, if the variable does follow a normal distribution, it cannot also have a tail skewed to the right. Are the mean and standard deviation the correct

descriptive statistics to use, if the variable is skewed to the right?

Sentences removed and median and IQ ranges added to Table 2.

4. Page 19, Table 2: Report hypertension as follows: n (%), similar to Table 3.

Done

5. Page 20, Table 3: I feel it is not necessary to have #p=0.008. Usually, only $p < 0.01$, $p < 0.05$ or $p < 0.1$ are

used in the footnote.

The reporting of exact p value will aid researchers who want to use our data for inclusion in meta-analyses to

derive more precise estimates without accessing raw data. If acceptable we would prefer reporting, where

suitable exact p values.

6. Page 20, Table 3: To what does "vs men" in the footnote refer to?

The statistical test refers to the comparison between men and women, testing sex differences. We have now

rephrased it to make it clearer.

2

7. Page 21, Table 4: To what does "vs men" in the footnote refer to?

See above

8. Page 22, Table 5: I feel it is not necessary to have #p=0.002. Usually, only $p < 0.01$, $p < 0.05$ or $p < 0.1$ are used in the footnote.

See answer to #5.

Reviewer: 3

Overall comments:

Given WHO's aim for member states to reduce salt consumption by 30% by 2025 in order to improve health outcomes at a population level in the most cost-effective way, this paper is highly relevant and will inform salt reduction strategies in Oman. However, the methodology currently lacks some specific details – which can be found in comprehensive supplementary file, however I think would be beneficial to be included in the actual paper in brief form. The discussion section could also be expanded and the findings of potassium included, along with acknowledging the importance of the Na:K. Please see comments below.

Abstract:

1. Objectives could state who the Na, K intake, KABS were measured in whilst this is mentioned in the setting

I think the objective could be more specific with this detail– i.e. similar to the title – To estimate population.....in adults living in Oman

We have amended the Objectives and changed the Title as suggested.

2. Participants: Seems to be a typo – no space between women 18 years or older. Suggest replacing “They

attended a” To “Participants attended a...”

Amended as suggested

3. Primary and secondary outcome measures: suggest include a word to describe the questionnaire – i.e. hard

copy questionnaire or online questionnaire

Amended as suggested

4. Results: Were these values adjusted for non-urinary loss of sodium? Did men eat significantly more Na and

K than women? It's unclear.

The estimation of salt and potassium intake from 24h urinary excretions were adjusted for non-urinary losses as

detailed in the Methods section. Men ate more salt and potassium than women and we have re-phrased the

sentence to make it clearer.

5. Conclusion: suggest replacing “back up” with “support” or similar

Amended as suggested

Strengths and limitations

1. Is the study nationally representative? How? By age and gender? N=569 (193 M, 376F) – is this representative of Oman? What about the spread of age?

The present study used the sampling frame of the 2017 WHO STEPS Surveys carried out in Oman. For the

purpose of the present salt and potassium survey a stratified random sample was obtained from the original

WHO STEPS Survey.

https://www.who.int/ncds/surveillance/steps/Oman_STEPS_2017_Data_Book.pdf?ua=1

https://www.who.int/ncds/surveillance/steps/Oman_STEPS_2017_Fact_Sheet.pdf?ua=1

We have now added a comparative table in the Supplementary documents relabelled as Table S1.

3

2. What do you mean by adopted a stringent quality control process to minimise the use of incomplete urine

collections? I understand word constraints here, but more succinct detail on these quality control processes

would be helpful.

The quality control protocol is explained in great detail in the Methods and the flowchart of Figure 1 gives the details. Difficult to list all the different steps in one sentence.

3. The last point about selection bias and limitations with generalisability seems to work against point 2 which

states this study population is a nationally representative cohort? This is confusing for the reader.

What do

you mean here?

Following also the Editorial requests we have now rephrased the section on Strengths and Limitations, using

only 5 bullet points with a sentence each addressing issues of methodology. The sample reflects the population

of Oman for general characteristics; of those of were selected 43% were excluded due to quality control on

urine collections. Their characteristics did not differ from those of the participants who provided valid collections. Nevertheless, it is difficult to completely rule out the presence of some sort of selection bias.

Introduction

1. Line 46 – how is the habit of eating out/increasing in urban areas contributing to salt consumption?

You

could expand here a little.

We have re-phrased the sentence reflecting the local knowledge and understanding. Oman has a striking

difference between the few urbanized areas of the country where eating out is the norm and the rest of the

country still relying of subsistence economy.

2. Line 48: what are the current salt reduction strategies – can you briefly describe these to give the reader an

idea of how your work contributes to them?

Re-phrased to clarify that the main contribution of the present study is in the monitoring and surveillance prong

of a three-pronged strategy of Communication, Reformulation, Monitoring & Surveillance.

3. Line 52: I think you could also discuss the importance of Na:K ratio and benefits for health here, not just

focusing solely on K?

We agree that the shift of the sodium-to-potassium ratio is the ultimate aim to maximize cardiovascular benefits.

We have considered this point hard enough and we concluded that the current global recommendations of

reducing salt consumption and at the same time increasing potassium consumption towards a 90 mmol/day target

would in itself deliver a change in the sodium-to-potassium ratio. Expanding would create confusion in public

health messages. In fact, currently the only place where it has become a point of further discussion is in China,

where the strategy appears to be a salt replacement strategy, not on the table in Oman and neighbouring

countries. We would therefore rather not expand too much on the Na/K as an indicator, not included in WHO

Guidelines for now.

Materials and methods

1. Data collection Line 8: What was the rationale for the participant presenting with a full bladder 30 minutes before measurements?

The instruction aims at avoiding that too much residual urine is left in the bladder and that the participant is unable to void when asked due to the previous voiding. This would contribute to the risk of underestimating the urine collection.

2. Data collection line 22: did you collect any medications participants might regularly take? Some medications were collected. For Hypertension we only had information on Yes/No on therapy, used to define hypertension in addition to the BP cut off points.

4

3. Data collection line 54: the following day after the 24 hour collection, did field team-members visit households for the urine collection, or did the participant take the urine collection somewhere? Field team-members visited the households again; text modified to make this clear.

4. Line 31 – you mention the modules that were part of the survey, but how many questions did it include?

Was the questionnaire a hard copy or asked verbally? How long did it take the participant to complete the

survey? Given the focus in this paper is on KABs, what were the specific survey questions related to KABs? More detail here would be helpful for the reader. You present something similar in Line 43 of the

results which might be helpful to be moved here: Knowledge, attitude and behaviours toward the consumption of salt was assessed by asking participants about the frequency, quantity and type of salt used

in the household, as well as their cooking habits and their attitudes towards dietary salt

We cannot identify specifically Line 31 – however, we answered the questions raised as follows: the specific

tool is the country adaptation of the WHO STEPS, referenced as Ref #26. The questionnaire consisted of 420

questions and it was administered verbally by field team members through an electronic tablet with embedded

software. This has been added to the revision. The questions specifically related to KAB are listed in Table 4. A

section has been shifted here and modified to indicate the specific survey questions related to KABs on dietary salt intake.

5. Line 60 onwards – “we either carried out determinations immediately or stored samples in the fridge until

the determination (as soon as possible). We determined sodium, potassium, creatinine immediately” – this

second sentence is opposite to the first? This is confusing for the reader.

Apologies – agree with the reviewer. We have now removed the second sentence.

6. Statistical analysis – I think sample size calculations would be better suited to be presented early – participants per stratum etc – this has not been clearly explained from the outset which is important and

relates to your recruitment process earlier described.

We have moved the section describing sample size calculations to the earlier section describing Participants

and Recruitment, as suggested by the reviewer.

7. Strengths and limitations dot point stated that “the study has adopted a stringent quality control process to minimize the use of incomplete urine collections” – where is the description of these processes in the methods section?

The quality control process is described on page 6 of 42, lines 34-44 and shown as Flowchart in Figure 1.

8. Did you adjust your sample with any post-stratification weights?

We did not apply post-stratification weights.

9. Was the length of the urine collection standardised to a 24h period?

Collections of less than 23h or more than 25h were excluded. Those in this range were standardised to 24h.

Now clarified in the text.

10. Was creatinine used as a measure of urine collection completeness?

The urinary excretion of creatinine in the 24h was one of the criteria to assess completeness. Any value outside

2 standard deviations on either side of the sex-specific frequency distribution were excluded, as stated in Figure

1.

11. What were the exclusion criteria for incomplete urines? E.g. creatinine levels by gender, outliers, missing urine etc.

Again, Figure 1 shows the criteria: Missing more than one void, duration less than 23h and more than 25h,

volumes less than 500ml/24h and urinary creatinine outside 2 standard deviations of sex-specific distributions.

5

Results

1. Line 58 – how did you define processed foods? As per NOVA classification, or otherwise? This should be

stated in your methods section when describing the survey questions

The definition of processed food was by WHO STEPS protocol with localised cards. A definition of this term has

now be inserted in the Methods section under Data Collection.

2. Were there any differences in consumption by socioeconomic status?

We did not look at any difference as the sample size would not have allowed us to rule out the risk of false

negatives (type I error) in statistical inference.

Discussion

1. The health vision is mentioned in line 56 – given this I think it would be helpful to include some context

about what this is here, or in the introduction when you discuss salt reduction interventions.

This is a good suggestion. Thank you. We have added a paragraph to explain the context

2. Reasons for differences in salt /potassium consumption by governorates? Do they differ in rurality? Are

there differences in local dishes consumed? Any insight the authors can provide to interpret these findings?

Whilst there might be small variations in recipes from place to place, the overriding factor would be greater

access and use of manufactured, processed food consumed in urban areas (Muscat, Dhofar, North Batinah)

compared to rural areas.

3. RE: comparison with other countries (GCC and Arab Peninsula) – some specific comparison of your study findings compared to other countries would be helpful. Do these countries currently have health visions/salt reduction strategies which might influence population salt consumption? We do mention specifically on page 11 of 42 (lines 10-22) some countries for comparison (Bahrain S Arabia, Qatar, Lebanon and Jordan). The EMRO Region of WHO has a salt reduction strategy and many countries have pledged the implementation of such strategy. However, actions are slow and vary from country to country. We have quoted Ref #18 and have now added a paragraph to summarise the state of progress in the EMRO and Gulf regions.

4. Given the focus was also KABs, this seems to be missing from the discussion section – how does this compare to similar nations? This would also feed into informing the health vision/direction of intervention strategies? We do make reference to neighbouring countries with similar habits and comparable traditions as far as bread making and consumption later in Discussion, when comparing results with GCC and Arab Peninsula.

5. Strengths and Limitations: line 36 – how is including all adults a strength of this study? It is a population based survey, but how reflective is your sample by age/gender to national population data? We have added a comparative table (Table S1) in Supplementary material to address this question, also raised elsewhere when referring to national representativeness.

6. Line 43: Fifth, we applied a rigorous quality control protocol to ensure completeness of urine collections, and to minimize both under and over-estimations.....there is no detail described about what these processes/protocol was. We have clarified this point in previous answers in Methods

7. Limitations: First, we analysed only 57% of the urine samples originally collected from willing individuals. This was due to the stringent quality control that has led to the exclusion of incomplete or erroneous collections.....I thought 57% was the response rate – i.e. number of people consented/ number people invited...is this figure now only the proportion of complete urines collected? This is a little confusing and would benefit from clarification throughout the paper. And how does this introduce self-selection bias? Can you provide a rationale for this? Thank you. We agree that the terminology may be confusing. We have now added Table S1 in the supplementary table to address the national representativeness. The sub-sample we took for the salt and potassium survey is then described in detail in our paper. All people selected were visited. The response rate of 57% is the combination of incomplete urinary data collection (missing data) and the quality control on completeness of urine collection. In a handful of cases participants with some missing data (height, weight, blood pressure) were

still included due to complete urine collections, since our primary objective was to assess salt and potassium consumption (now stated in footnote of Table 3).

8. A potential limitation is bias/changing one's diet prior to the urine collection – whilst acknowledged in the methods section as an instruction not to change diet, this could still be a limitation and worth mentioning here.

Added

9. Can you acknowledge that without knowing dietary intake, i.e. kJ E consumed in males and females, that males may have higher sodium content due to consuming more food?

We agree with the reviewer that the sex difference is almost entirely due to the fact that men eat more food than women because they are bigger (taller and heavier), even with comparable BMI. This is a consistent finding of surveys of this kind in different countries and continents. We have added a sentence as we have always done in our previous surveys, place in the Results section.

10. Without really understanding how the questionnaire was administered (i.e. by an interviewer? On paper?

Online?) It is hard to know whether social desirability bias (to the interviewer) or recall bias could influence responses.

WHO Steps questionnaire were administered by field workers (as explained earlier and incorporated in revised text). Suspecting biases is legitimate by the reviewer and they are difficult to remove altogether. Any suggestion would be welcome.

11. I think this data is limited in some respects to informing intervention strategies as diet recalls have not been conducted therefore key sources of dietary salt and potassium aren't known.

We agree with the reviewer. We have clearly stated this in limitations (page 12 of 42, lines 25-30)

12. RE: policy implications "These indications, together with the awareness and behaviours measured, suggest that policy priorities⁴⁵ to reduce population salt consumption in the Sultanate of Oman would require (a)

improvement of salt-related knowledge through health promotion campaigns, (b) assessment of major sources of salt consumption, (c) establishing collaborations with the local authorities to reduce the amount

of salt used in traditional bread making and other identified sources like locally produced condiments, (d)

adopting a labelling strategy for imported foods with high salt content." Can you explain what you mean

by policy priorities? What sort of policy do you propose might be promising in improving salt related KABs? How will policies assess major sources of salt consumption? Can the authors expand upon this,

and draw from existing salt reduction strategies?

We have re-phrased this paragraph to make it clear that we refer to initiatives to be taken to reduce consumption.

13. Bigger picture implications of this data could be expanded – could this data inform salt reduction strategies?

We have now re-phrased the paragraph. We do identify areas for action. Salt is high, potassium is low.

Awareness is limited and, from a variety of neighbouring countries and local data, bread seems to be the first

target for reformulation. Indeed, some local studies are in progress with bakeries (see also revised Conclusions).

7

14. Potassium levels are not addressed in the discussion – how do you propose you increase potassium

consumption to adequate levels? And what about the importance of the Na:K?

We agree that the shift of the sodium-to-potassium ratio is the ultimate aim to maximize cardiovascular benefits.

We have considered this point hard enough and we concluded that the current global recommendations of

reducing salt consumption and at the same time increasing potassium consumption towards a 90 mmol/day target

would in itself deliver a change in the sodium-to-potassium ratio. Expanding would create confusion in public

health messages. In fact, currently the only place where it has become a point of further discussion is in China,

where the strategy appears to be a salt replacement strategy, not on the table in Oman and neighbouring

countries. We would therefore rather not expand too much on the Na/K as an indicator, not included in WHO

Guidelines for now.

Conclusion:

1. lines 31. These are great suggestions for public health promotion, however the data in your study does not

appear to directly inform these approaches?

We have now re-phrased the paragraph. We do identify areas for action. Salt is high, potassium is low.

Awareness is limited and, from a variety of neighbouring countries and local data, bread seems to be the first

target for reformulation. Indeed, some local studies are in progress with bakeries.

2. Table 2: missing a key/legend for *

Apologies – missing legend for significance test between men and women – now amended.

3. Table 3: * $p < 0.05$ – can you clarify that this analysis was conducted on males vs females? What does

+ $p < 0.001$ vs men mean?

The statistical test refers to the comparison between men and women, testing sex differences. We have now

rephrased it to make it clearer.

4. Table 4: * $p < 0.05$ – can you clarify that this analysis was conducted on males vs females? What does

+ $p = 0.04$ vs men mean? RE: reduced numbers due to missing values – how many were missing values and

why were they missing? Did you conduct an analysis with just complete values and were the same % /

effects seen? Same comments for Table 5.

The statistical test refers to the comparison between men and women, testing sex differences. We have now

rephrased it to make it clearer. We have added missing numbers in Table. Analysis performed on maximum numbers available. No difference in effect seen in sensitivity analysis with only complete data.

VERSION 2 – REVIEW

REVIEWER	Dr Marike Cockeran North-West University South Africa
REVIEW RETURNED	15-Jun-2020

GENERAL COMMENTS	I have reviewed the statistical aspects of the manuscript. Independent t-tests and Pearson's chi-square tests were performed. It seems that the correct analyses are performed and the results appropriately reported.
--

REVIEWER	Kristy A Bolton Deakin University, Australia
REVIEW RETURNED	18-Jun-2020

GENERAL COMMENTS	Thank you for strengthening your draft. My key concern is related to the Discussion section. Discussion Original reviewer query: Potassium levels are not addressed in the discussion – how do you propose you increase potassium consumption to adequate levels? And what about the importance of the Na:K? Author response: We agree that the shift of the sodium-to-potassium ratio is the ultimate aim to maximize cardiovascular benefits. We have considered this point hard enough and we concluded that the current global recommendations of reducing salt consumption and at the same time increasing potassium consumption towards a 90 mmol/day target would in itself deliver a change in the sodium-to-potassium ratio. Expanding would create confusion in public health messages. In fact, currently the only place where it has become a point of further discussion is in China, where the strategy appears to be a salt replacement strategy, not on the table in Oman and neighbouring countries. We would therefore rather not expand too much on the Na/K as an indicator, not included in WHO Guidelines for now. Reviewer rebuttal: Given that you have potassium in your title, it is a key objective of your study, and you present findings on potassium consumption/excretion, you need to compare/contrast this to other studies and interpret your findings/present implications of this new knowledge in the discussion. There is currently nothing about potassium in your discussion section. Other minor queries: Introduction Original reviewer query: Line 52: I think you could also discuss the importance of Na:K ratio and benefits for health here, not just focusing solely on K? Author response: We agree that the shift of the sodium-to-potassium ratio is the ultimate aim to maximize cardiovascular
---

benefits. We have considered this point hard enough and we concluded that the current global recommendations of reducing salt consumption and at the same time increasing potassium consumption towards a 90 mmol/day target would in itself deliver a change in the sodium-to-potassium ratio. Expanding would create confusion in public health messages. In fact, currently the only place where it has become a point of further discussion is in China, where the strategy appears to be a salt replacement strategy, not on the table in Oman and neighbouring countries. We would therefore rather not expand too much on the Na/K as an indicator, not included in WHO Guidelines for now.

Reviewer rebuttal: I disagree that discussion of the importance of Na:K would create confusion about public health messages. The message isn't to increase sodium consumption to meet potassium levels, and this certainly that isn't what the published literature is saying. This point is to acknowledge that potassium shouldn't only be examined in isolation, with recent evidence suggesting that a high Na:K is more strongly related to CVD risk compared to sodium or potassium in isolation (reference Weaver, C.M. Potassium and health. *Adv. Nutr.* 2013, 4, 368S–377S, Cook, N.R.; Obarzanek, E.; Cutler, J.A.; Buring, J.E.; Rexrode, K.M.; Kumanyika, S.K.; Appel, L.J.; Whelton, P.K. Trials of Hypertension Prevention Collaborative Research, G. Joint effects of sodium and potassium intake on subsequent cardiovascular disease: The Trials of Hypertension Prevention follow-up study. *Arch. Intern. Med.* 2009, 169, 32–40). It just adds to the case of importance of potassium.

Materials and methods

Original reviewer query: Data collection Line 8: What was the rationale for the participant presenting with a full bladder 30 minutes before measurements?

Author response: The instruction aims at avoiding that too much residual urine is left in the bladder and that the participant is unable to void when asked due to the previous voiding. This would contribute to the risk of underestimating the urine collection.

Reviewer rebuttal: I think adding some text to justify this in your manuscript would be beneficial. E.g. presenting with a full bladder 30 minutes before measurements to reduce the risk of underestimating urine collection.

Original reviewer query: Data collection line 22: did you collect any medications participants might regularly take?

Author response: Some medications were collected. For Hypertension we only had information on Yes/No on therapy, used to define hypertension in addition to the BP cut off points.

Reviewer rebuttal: I think adding text to acknowledge data on medications was collected in your methods is helpful for the reader.

VERSION 2 – AUTHOR RESPONSE

Reviewer: 2

Q. I have reviewed the statistical aspects of the manuscript. Independent t-tests and Pearson's chi-square tests were performed. It seems that the correct analyses are performed and the results appropriately reported.

A. *Thank you*

Reviewer: 3

Thank you for strengthening your draft. My key concern is related to the Discussion section.

Discussion

Original reviewer query: Potassium levels are not addressed in the discussion – how do you propose you increase potassium consumption to adequate levels? And what about the importance of the Na:K?

Author response: We agree that the shift of the sodium-to-potassium ratio is the ultimate aim to maximize cardiovascular benefits. We have considered this point hard enough and we concluded that the current global recommendations of reducing salt consumption and at the same time increasing potassium consumption towards a 90 mmol/day target would in itself deliver a change in the sodium-to-potassium ratio. Expanding would create confusion in public health messages. In fact, currently the only place where it has become a point of further discussion is in China, where the strategy appears to be a salt replacement strategy, not on the table in Oman and neighbouring countries. We would therefore rather not expand too much on the Na/K as an indicator, not included in WHO Guidelines for now.

Reviewer rebuttal: Given that you have potassium in your title, it is a key objective of your study, and you present findings on potassium consumption/excretion, you need to compare/contrast this to other studies and interpret your findings/present implications of this new knowledge in the discussion. There is currently nothing about potassium in your discussion section.

A. *We have added in Discussion comparisons with levels of potassium intake in other countries, as suggested.*

Other minor queries:

Introduction

Original reviewer query: Line 52: I think you could also discuss the importance of Na:K ratio and benefits for health here, not just focusing solely on K?

Author response: We agree that the shift of the sodium-to-potassium ratio is the ultimate aim to maximize cardiovascular benefits. We have considered this point hard enough and we concluded that the current global recommendations of reducing salt consumption and at the same time increasing potassium consumption towards a 90 mmol/day target would in itself deliver a change in the sodium-to-potassium ratio. Expanding would create confusion in public health messages. In fact, currently the only place where it has become a point of further discussion is in China, where the strategy appears to be a salt replacement strategy, not on the table in Oman and neighbouring countries. We would therefore rather not expand too much on the Na/K as an indicator, not included in WHO Guidelines for now.

Reviewer rebuttal: I disagree that discussion of the importance of Na:K would create confusion about public health messages. The message isn't to increase sodium consumption to meet potassium levels, and this certainly that isn't what the published literature is saying. This point is to acknowledge that potassium shouldn't only be examined in isolation, with recent evidence suggesting that a high Na:K is more strongly related to CVD risk compared to sodium or potassium in isolation (reference Weaver, C.M. Potassium and health. *Adv. Nutr.* 2013, 4, 368S–377S, Cook, N.R.; Obarzanek, E.; Cutler, J.A.; Buring, J.E.; Rexrode, K.M.; Kumanyika, S.K.; Appel, L.J.; Whelton, P.K. Trials of Hypertension Prevention Collaborative Research, G. Joint effects of sodium and potassium intake on subsequent cardiovascular disease: The Trials of Hypertension Prevention follow-up study. *Arch. Intern. Med.* 2009, 169, 32–40). It just adds to the case of importance of potassium.

A. *We have now added Na:K in Table 3. We agree with the reviewer that observational studies point to an association between Na:K ratio and CVD risk. However, observational studies do not provide an indication to set guidelines of what a target ratio should be. On the other hand, the use of Na/K ratio is helpful when monitoring changes in intervention trials or surveillance. We now discuss also the implications for Na:K and add several references.*

Materials and methods

Original reviewer query: Data collection Line 8: What was the rationale for the participant presenting with a full bladder 30 minutes before measurements?

Author response: The instruction aims at avoiding that too much residual urine is left in the bladder and that the participant is unable to void when asked due to the previous voiding. This would contribute to the risk of underestimating the urine collection.

Reviewer rebuttal: I think adding some text to justify this in your manuscript would be beneficial. E.g. presenting with a full bladder 30 minutes before measurements to reduce the risk of underestimating urine collection.

A. *Sentence amended as suggested*

Original reviewer query: Data collection line 22: did you collect any medications participants might regularly take?

Author response: Some medications were collected. For Hypertension we only had information on Yes/No on therapy, used to define hypertension in addition to the BP cut off points.

Reviewer rebuttal: I think adding text to acknowledge data on medications was collected in your methods is helpful for the reader.

A. *Number and frequency on anti-hypertensive medications (yes or no) added in Table 2.*

VERSION 3 – REVIEW

REVIEWER	Kristy A Bolton Deakin University, Australia
REVIEW RETURNED	27-Aug-2020
GENERAL COMMENTS	Best of luck with future research.